# Symptoms of Anxiety, Depression, and Post-Traumatic Stress Disorder in Health Care Personnel in Norwegian ICUs during the First Wave of the COVID-19 Pandemic, a Prospective, Observational Cross-Sectional Study

**DOI:** 10.3390/ijerph19127010

**Published:** 2022-06-08

**Authors:** Siv Karlsson Stafseth, Laila Skogstad, Johan Ræder, Ingvild Strand Hovland, Haakon Hovde, Øivind Ekeberg, Irene Lie

**Affiliations:** 1Department of Postoperative and Intensive Care, Division of Emergencies and Critical Care, Oslo University Hospital, 0424 Oslo, Norway; 2MEVU, Lovisenberg Diaconal University College, 0456 Oslo, Norway; 3Department of Health and Care Sciences, Faculty of Health Sciences, UiT The Arctic University of Norway, 9019 Tromsø, Norway; laila.skogstad@uit.no; 4Department of Anaesthesiology, Division of Emergencies and Critical Care, Oslo University Hospital, 0424 Oslo, Norway; johan.rader@medisin.uio.no; 5Institute of Clinical Medicine, University of Oslo, 0313 Oslo, Norway; 6Department of Acute Medicine, Division of Medicine, Oslo University Hospital, 0424 Oslo, Norway; istran@ous-hf.no; 7The Norwegian Association for Critical Care Nurses, 0152 Oslo, Norway; haakonhovde@hotmail.com; 8Psychosomatic and Consultation-Liaison Psychiatry, Division of Mental Health and Addiction, Oslo University Hospital, 0424 Oslo, Norway; oeekeber@online.no; 9Centre for Patient-Centered Heart and Lung Research, Department of Cardiothoracic Surgery, Division of Cardiovascular and Pulmonary Diseases, Oslo University Hospital, 0424 Oslo, Norway; uxirli@ous-hf.no; 10Department of Health Sciences in Gjøvik, Faculty of Medicine and Health Sciences, Norwegian University of Science and Technology, 2815 Gjøvik, Norway

**Keywords:** COVID-19, intensive care units, health personnel, anxiety, depression, stress disorder, PTSD, social interaction

## Abstract

Background: The COVID-19 pandemic has induced demanding work situations in intensive care units (ICU). The objective of our study was to survey psychological reactions, the disturbance of social life, work effort, and support in ICU nurses, physicians, and leaders. Methods: From May to July 2020, this cross-sectional study included 484 ICU professionals from 27 hospitals throughout Norway. Symptoms of anxiety and depression were measured on Hopkins Symptom Checklist-10 (HSCL-10). Symptoms of post-traumatic stress disorder (PTSD) were measured on the PCL-5. Results: The study population were highly educated and experienced professionals, well prepared for working with COVID-ICU patients. However, 53% felt socially isolated and 67% reported a fear of infecting others. Probable cases of anxiety and depression were found in 12.5% of the registered nurses, 11.6% of the physicians, and 4.1% of the leaders. Younger age and <5 years previous work experiences were predictors for high HSCL-10 scores. Reported symptom-defined PTSD for nurses 7.1%; the leaders, 4.1%; and 2.3% of physicians. Conclusions: ICU health care professionals experienced talking with colleagues as the most helpful source of support. The COVID-ICU leaders reported a significantly higher mean score than physicians and nurses in terms of pushing themselves toward producing high work effort.

## 1. Introduction

In March 2020, The World Health Organization (WHO) declared that the 2019 coronavirus (with COVID-19) outbreak had reached the level of a pandemic, due to a rapid worldwide spread with a significant threat to health [1]. In Europe, at the end of the first pandemic wave in June 2020, 1.6 million people had contracted COVID-19, with more than 100,000 deaths [2]. In the same period in Norway, 9000 cases of COVID-19 were identified, with 217 admissions reported into intensive care units (ICUs) and a mortality rate of 18.4% (n = 40) [3,4]. In Norway, the first wave of the pandemic was characterized by geographic variation in cases. The south-eastern region Norway had large outbreaks while other regions saw few cases. The government implemented restrictions and stringent lockdown, i.e., to stay in municipality of residence. People were encouraged to work from their home office and to minimize social contact outside the family. Schools were closed, and hospital visits were restricted. It was necessary to “build” COVID-ICUs and wards to special care for the patients were created. Personnel were asked to work in new areas and wards, also with lack of personal protective equipment and increased risk of infection [3].

A key question is to what extent this first wave will affect the psychological reactions, the disturbance of social life, work effort, and support in health care professionals (HCPs): ICU nurses, physicians and leaders.

### 1.1. Prevalence of Psychological Reactions

Worldwide, the health care system and HCPs in COVID-ICUs experienced a massive psychological burden, especially in Italy [5] and China [6,7]. Studies across the world since the COVID-19 pandemic outbreak confirm that HCPs have experienced psychological distress [8,9] in symptoms of anxiety and depression (mental health disorder) and post-traumatic stress disorder (PTSD) [10,11]. ICU-HCPs working in greater epidemic intensity, based on public data for the region, experienced significantly higher risk of mental health and overall perceive distress compared to HCPs working in low-intensity regions as seen in Laurent et al.’s study including 77 French hospitals (n = 2643) [12]. Several studies and reviews have confirmed that HCPs working with COVID-19 patients experience symptoms of anxiety, depression, and symptom-defined post-traumatic stress disorder (PTSD) [9,12,13,14,15]. In a rapid systematic review including 22 studies of different HCP settings, anxiety ranged from 9% to 90%, with a median of 24% [8]. Moreover, depression reported from 19 studies ranged from 5% to 51%, with a median of 21% [8]. Studies on psychological reactions among HCPs (nurses and doctors) working in critical care settings with COVID-19 patients have been performed in Europe (France, United Kingdom, Ireland), Canada, Australia, and New Zealand. The studies have different ranges of samples (n = 109–2955) and questionnaires. They found prevalence of the following in the symptoms of HCPs: anxiety in 27–67%; depression in 16–57%; and PTSD in 22–65% [13,14,15,16,17,18,19]. In a rapid review of the mental health impact of COVID-19 on HCPs, there were data on sleep problems in six observational studies (n = 9105) and such problems were reported in a median of 37% (range 34–65) [8], while 68% of Iraqi physicians from different medical settings experienced sleeplessness [20]. 

Managers and head (leaders) of ICU-departments are, according to World Health Organization (WHO), facing similar forms of stress as the HCPs working bedside them. In addition, they may experience pressure relating to the responsibilities of their role [21]. 

### 1.2. Factors Associated with Psychological Reactions

In a French study of critical care clinicians in 21 ICUs (n = 1058), six independent factors were associated with psychological reactions: fear of being infected; inability to rest; inability to care for family; struggling with difficult emotions; regret about the restrictions in visitation policies; and witnessing hasty end-of-life decisions [15]. Moreover, studies have shown an increase in psychological reactions in women [7,22], being a nurse [23], and worrying about self or family members being infected [7,16,24,25]. Other significant factors were anxiety with respect to the availability of personal protective equipment (PPE) [16,26], being responsible for other staff members, being asked to work outside one’s expertise area and not feeling supported in the work environment and team [13]. Moreover, worldwide, the country was under the government’s decision to follow a national “lock-down” of society during the first wave of the pandemic (May–July 2020) to restrict the spread of the virus. This implied a reduction in social contact and meeting people, wearing masks, and physical distance resulting in disturbance of social life [3].

At work, HCPs could experience a lower level of work satisfaction, or the opposite if management appreciated the individual efforts in the care of COVID-19 patients [27]. Kramer et al. studied German HCPs (n = 3669) and found nurses reported higher levels of stress and increased subjective burden of workload compared to physicians [27]. Research in relation to turnover and intention to leave the profession remain sparse, but one Canadian study found 22.3% of front-line nurses considered quitting their job [19]. 

### 1.3. Support

Social support is recognized as an important protective factor to reduce mental health problems [8,26]. In a study, during spring 2020, at a large medical centre in New York the HCPs (n = 657) reported talk therapy (26%) and virtual provider support groups (16%) as the preferred social support to cope with psychological symptoms [28]. Moreover, HCPs experienced an increased sense of meaning and purpose (61%) in their work since the COVID-19 outbreak [28]. At the same time in Norway, 65% of HCPs experienced feeling lonely [25]. In a Swedish hospital, they rapidly implemented a comprehensive psychological support model for ICU frontline workers and support for managers, during the first wave of COVID-19 [29]. Peer support initiatives, as well as daily group sessions during work hours, were the foremost preferred psychological support for HCPs reported in the process evaluation of the model [29]. In a US nationwide study, critical care nurses (n = 285) reported inadequate leadership support as an important stress factor [30].

Even though there are several studies on psychological distress among health care professionals, there is a lack of studies from Scandinavia and especially including leaders working in COVID-ICUs. 

Our research aims were to survey symptoms of anxiety, depression, and PTSD in the sub-groups of nurses, physicians, and leaders in COVID-ICUs. Moreover, we aimed to survey satisfaction with the respondents’ own professional work effort, disturbance of social life, and support measures. This report represents part II of a national COVID-19 intensive care study focusing on both data on physical preparedness and working conditions [25], and psychological reactions with the following objectives:What was the prevalence of probable clinical cases of anxiety, depression, and symptom-defined PTSD among COVID-ICU health care professionals, i.e., nurses, physicians, and leaders in Norway?What demographic characteristics were associated with the probable mental health disorder in COVID-ICU health care professionals?To what extent did COVID-ICU health care professionals in Norway suffer from disturbance of their social life during the first wave of the pandemic?How did COVID-ICU health care professionals in Norway self-assess their work effort, and what kind of support was received and perceived as useful?

## 2. Materials and Methods

### 2.1. Design, Setting, and Recruitment Procedure

This observational cross-sectional study investigated self-reported psychosocial reactions and support measures in a nationwide cohort of ICU health care personnel in Norway during the first wave of the COVID-19 pandemic in 2020. Respondents were defined as frontline HCPs actually caring for COVID-19 patients in centralized 28 hospitals, satisfying the criterion of having a COVID-ICU. Nurses, physicians, and their leaders in 28 were invited to participate when their local leader redistributed the invitation including general information about the study with a direct link available for voluntary participation from 6 May to 14 July 2020. The HCPs could voluntarily follow a direct link for electronic signature to the informed consent, and could continue to respond the study questionnaire. The survey was administered online through the University Information Technology Center (USIT) at University of Oslo (https://www.uio.no/english/services/it/adm-services/nettskjema/about-nettskjema.html, accessed on 1 May 2020).

### 2.2. The Study Questionnaire

The baseline study questionnaire was a composite package of 181 questions divided into eight parts. A previously published paper presents the ICU-HCPs’ preparedness and working conditions [25]. For the present paper, to focus on psychological reactions and support measures, 69 items were relevant as shown in the Appendix A. The questionnaire included two validated checklists: the Hopkins Symptom Checklist-10 (HSCL-10) and the PTSD checklist for the Diagnostic Statistical Manual of Mental Disorders (DSM-5) in PCL-5 [10,11]. Moreover, the authors of this paper, e.g., COVID-ICU clinicians and researchers, developed questions for the study questionnaire by using a modified Delphi method, i.e., a group consensus strategy that uses literature review and clinical experiences from interdisciplinary experts in the development of relevant questions [25,31].

### 2.3. Checklists for Psychological Symptoms

Current symptoms of anxiety and/or depression and probable symptom-defined PTSD were assessed with two international recognized checklists, previously translated, and validated into Norwegian [10,11]:The Hopkins Symptom Checklist-10 (HSCL-10) measures 10 items of anxiety- and depression-related symptoms on a four-point Likert scale (1 = Not bothered to 4 = Very much bothered) [21]. Four items represent symptoms of anxiety (i.e., fearful, afraid, dizzy and feeling tension), while the other six items (i.e., claims, sleep disturbance, depression, useless, struggle, and hopelessness) represent symptoms of depression. A cut-off value of ≥1.85 on the mean value of 10 points was set for a probable mental health disorder with a previously reported sensitivity of 89% and a specificity of 98% [10]. The time window for recapitalizing memories was “last week”, as recommended in earlier studies [28,32]. In the present study, the Cronbach’s alpha was 0.89.The PTSD checklist for DSM-5 (PCL-5) is a 20-item self-report measure that examines symptoms of PTSD using 5-point Likert scale (0 = Not at all to 4 = Extremely) ‘during the last month’ [11,32]. In the present study, the symptoms are related to working in an ICU with COVID-19 patients. The scale has a range of 0–80, and a cut-off score of ≥31 has been reported to identify cases of PTSD with clinical significance [33,34,35,36]. In the present study, the Cronbach’s alpha was 0.93.

### 2.4. Study Developed Questions: Individual Work Effort and Support Measures

The assessment of individual work effort, support measures from the employer, professional satisfaction and proficiency during work, recognition and support during the pandemic, and appreciation of effort from hospital, family or friends and the responsible authorities were scored using a 5-point Likert scales (1 = not at all, 2 = to a small degree, 3 = partly, 4 = to a high degree, 5 = to a very high degree). In the present study, the Cronbach’s alpha of work effortwas 0.42 and for recognition and support 0.49. Then, a multiple-choice question regarding support from the employer during the COVID-19 effort gave the possibility to choose one or more of these options: no special support, group meetings (debriefing, defusing); professional support by a dedicated person (psychologist, psychiatric nurse, chaplain, etc.); follow-up by leader after completed shifts. There were also options of “support offered but declined” and “not applicable”.

Three questions focused on more personal areas: (1) disturbance on the respondent’s social life, fear of infecting family, colleagues, or patients; (2) social contacts outside working hours, use of social media (Facebook, Skype, etc.) and phone calls to colleagues. Both questions one and two were assessed with Yes/No or N/A (“not applicable”); (3) social contact in terms of meeting people outdoors or indoors at distance wearing a mask, or alternatively in violation with these recommendations from the Norwegian Institute of Public Health (NIPH) [3], were assessed using 5-point Likert scales (1 = not at all, 2 = to a small degree, 3 = partly, 4 = to a high degree, 5 = to a very high degree).

### 2.5. Demographics

Age, sex, marital status, name of the hospital, occupation (nurse, physician, or leader) and length of work experience were assessed [25]. One question addressed changes of personal economy (income) during the early stage of the pandemic. Two questions addressed alcohol use and psychoactive medications taken for short- or long-term use (for calming purposes, as a mood enhancer, and to sleep), had response alternatives of “weekly”, “daily”, “sporadic” or “N/A”. The COVID-ICU health professionals were also asked if they had considered quitting their job due to the pandemic, with the response options, “Yes: planning, sometimes, several times” or “No”. There were no missing data as all response fields in the online survey system were mandatory. 

### 2.6. Statistical Analyses

The data are presented as the mean, standard deviation (SD), range (min–max) or 95% confidence interval (CI), or a percentage, as appropriate. The significance level is set at 5% (*p* = 0.05). Analysis with the Kruskal–Wallis test was used to compare groups where we had skewed data. For the test of correlation coefficients, Spearman’s rho (significant at the 0.05 level, 2-tailed) was used for categorical variables. Quantile regression analyses for skewed original scores in dependent variables were performed [37]. Demographic, independent variables (i.e., age, sex, married or partnered, professional, previous work experience), were included and tested in the regression models: univariate and multivariate. The models’ explanation of HSCL-10 (dependent variable for cases of probable mental health disorder) and PCL-5 (dependent variable for symptom-defined cases of PTSD) were examined within the limitation of included number of cases. The IBM SPSS (Statistical Package for the Social Sciences, Armonk, NY, USA) Statistics version 27 was used to carry out all analyses.

### 2.7. Ethics

The study was conducted in accordance with the Declaration of Helsinki and approved by the Regional ethics committee (2020/136144) and the Data protection officer at Oslo University Hospital (20/09438). Moreover, at each hospital and each COVID-ICU in Norway, approvals for the study were obtained from the Data Protection Officer, the Head of Research and the leaders. Online signed informed consent, and data from the study questionnaire were stored on a secured server. Results from the study were processed without names or other recognizable information. A code connects the informants’ information to a code list. A link key (name and study ID) is stored separately on a secured server at Services for Sensitive Data at the University of Oslo. The respondents had the possibility of contacting the project manager for answers to any questions or for support of any kind. The study was registered in Clinicaltrials.gov, NCT04372056.

## 3. Results

### 3.1. Demographics and Work Experience

In total, 484 health care professionals, working in COVID-ICUs at 27 hospitals throughout Norway completed the survey (Table 1). One ICU, out of the twenty-eight contacted, did not participate due to delays and problems with local approval of participation. Most (81.2%) respondents were working in the south-eastern region of Norway and cared for 73.3% of ICU patients diagnosed with COVID-19 [4], while the remaining were based in western, central, and the northern parts of the country. The respondents were generally highly educated, trained, and professionally experienced (Table 1). Registered nurses (RNs) were the largest group by profession (n = 392, 81.0%). Overall, 80% of the RNs were females, and 86% were critical care nurses. The group of physicians consisted of 43 (8.9%) participants, of whom 32% were females. The group of leaders, represented by 49 (10.1%) participants, included both nurses and physicians, with 71% being females. Eight percent (n = 39) were temporary staff without previous ICU experiences. The RNs had a mean time of work experience in ICUs of 19.6 years and the physicians reported a mean of 17.4 years. Most respondents (81%) had previous work experience from performing similar tasks in isolation units, including wearing masks and personal protective equipment (PPE). Most (74.8%) of the respondents were married or had a partner. 

### 3.2. Symptoms of Anxiety, Depression, Probable Symptom-Defined PTSD

Anxiety symptoms, measured with the HSCL-10 during the COVID-ICU work, were present in 10.7% of RNs, 11.6% of physicians, and 4.1% of leaders, whereas the corresponding incidence of depression were 13.0% (RNs), 9.3% (physicians), and 4.1% (leaders). When symptoms of anxiety and depression, i.e., mental health disorder were merged, the incidence was 12.5% in RNs, 11.6% in physicians, and 4.1% in leaders (Table 2).

A quantile regression analysis (Table 3) indicated that younger age (coeff. −0.16, CI = −0.028–−0.003, *p* = 0.017), and limited previous work experiences (1–5 years, as opposed to >5 years, coeff. 0.4, CI = 0.0970–0.703, *p* = 0.018) were the only significant factors associated with high HSCL-10 scores.

RNs had a significantly higher prevalence of symptom-defined PTSD (7.1%) versus leaders (4.1%) and physicians (2.3%), (*p* < 0.001) (Table 2). The quantile regression analysis did not show any significant predictor for symptom-defined PTSD shown in the Appendix A. A strong correlation of 0.80 (*p* < 0.001) (Spearman’s rho) was found between respondents with high scores in both the HSCL-10 and the PCL-5 checklists.

### 3.3. COVID-19, Fear, and Social Disturbance

More than half of all the responding health care professionals (n = 326, 67.4%) reported fear of infecting others, especially family members in their own household. Moreover, many reported a fear of infecting other family members (n = 299, 61.8%), colleagues (n = 298, 61.6%), or patients (n = 294, 60.7%), but also friends (n = 170, 35%). Fifty-three percent (n = 257) felt socially isolated, whereas one out of three (n = 150) felt lonely during the first wave of the pandemic. Three out of four (n = 363) reported increased contact via social media, telephone, or at work. Outside working hours, social contact was challenged, but partly compensated by frequent use of social media (Facebook, Skype, etc.) (n = 324, 66.9%), video chats with family and friends (n = 334, 69%), and phone calls to colleagues (n = 168, 34.7%). Social contact was also maintained by meeting people outdoors (n = 356, 73.6%), or indoors with a mask, and physical distance (n = 175, 36.2%). Only 32 HCPs (6.6%) reported having social contact indoors in violation with the recommendations from the Norwegian Institute of Public Health. 

### 3.4. Assessment of Individual Work Effort and Support at Work

The work efforts for RNs, physicians, leaders, and subsequent professional satisfaction and proficiency are summarized in Table 4.

Three out of four (n = 367, data not shown) HCPs were expressed in recognition of the work effort (i.e., to a high or very high degree), and through recognition and in addition support from family or friends (Table 5). Moreover, the work was experienced as meaningful and Kruskal–Wallis test (after correction) revealed a significant difference between leaders (0.012) compared to physicians, and RNs (0.002) compared to physicians. In receiving sufficient advice and support, significant differences were detected between the leaders (0.03) and physicians and between RNs (*p* < 0.001) and physicians. The variable overextending, i.e., in a large effort, was expected from the team, managers or colleagues results showed significant differences between groups physicians (0.012) to RNs, versus leaders (*p* < 0.001).

Support at work: almost all respondents (93.6%) perceived talking with colleagues as the most supportive measure in the COVID-ICU setting and outside of work hours. Receipt of organized support measures was reported by 39.5% (n = 191), while 37% (n = 177) did not receive any such support measures during the pandemic. Support in group meetings (e.g., briefings) was reported by 30.4% (n = 147) and professional support (e.g., by a psychologist or psychiatric nurse) by 28.9% (n = 140). Support phone calls from the Department of Human Resources were reported by 5.4% (n = 26) and follow-ups from a manager after a completed shift by 4.8% (n = 23). In addition, 8% (n = 39) were offered supportive talks but declined to participate.

### 3.5. Economy, Alcohol, and Psychoactive Drugs

The personal economy was improved for 30.8% (n = 149), unchanged in 66.5% (n = 332), and worsened in 2.7% (n = 13) during the pandemic.

Alcohol consumptions were reported by the majority, “as usual” (39.7%, n = 192), whereas 25.2% (n = 122) reported “less” use, and 10% (n = 48) “more” use. For those reporting “more than usual”, the purpose of use was “calming”, “as a mood enhancer”, or “to sleep”. Very few of the respondents reported daily use of psychoactive drugs, such as sleep medications (1.8%, n = 9) or tranquilizers (0.6%, n = 3). Sporadic use was reported for sleep medications (9.9%, n = 48), antidepressants (1.0%, n = 5), and anxiolytic medications (0.4%, n = 2).

A majority (64.0%, n = 310) were determined to continue in their present job, while one out of four (26%) had considered quitting their job during the first wave of the pandemic as illustrated in Appendix A. 

## 4. Discussion

The main results of this prospective, observational cross-sectional study are that HCPs as both clinicians (nurses and physicians) and leaders in Norwegian COVID-ICUs were experiencing low symptom levels of anxiety, depression, and PTSD during the first wave of the pandemic (May–July 2020). In quantile regression analysis, younger age, and limited work experiences (<5 years) significantly increased the symptoms of anxiety and depression. The leaders reported that they extended their limits of work effort to a higher degree relative to physicians and nurses. HCPs perceived their individual work effort as satisfying, successful, and meaningful, which may have protected them against stress reactions [25]. The physicians had perceived sufficient advice and support to a higher degree relative to nurses and leaders, in addition to perceiving the work as meaningful. The foremost supportive measure for the HCPs was talking with ICU colleagues.

### 4.1. Symptoms of Anxiety, Depression, and Symptom-Defined PTSD

HCPs working in critical care settings across the world during COVID-19 report different symptom levels of psychological reactions [8,23,38], with a diversity of methodological quality of the studies [8]. In the present study, slightly more cases of anxiety and depression were found in RNs and physicians compared to the ICU leaders. Approximately one-sixth of HCPs reported previous symptoms of anxiety and depression (16.1% and 20.0%, respectively). Nurses and physicians working bedside are directly exposed to the patients’ suffering, which may influence psychological distress, as Greenberg et al. underline in their UK study (n = 709) [18]. In our study, to be a younger HCP was a comparable risk factor for symptoms, and the study from Metha et al. [39] could confirm the results; however, the risk factor of being a less experienced HCP was not confirmed in the Canadian study. Less experienced nurses were found by Kramer et al. [27] to experience psychological distress. 

Interestingly, in a general Dutch population study during the first four months of the pandemic, no increase in the prevalence of anxiety and depression symptoms was found, but emotional loneliness increased significantly [40]. The prevalence of anxiety and depression symptoms in the present study was low compared to other studies [8], but comparable to UK HCPs working in ICUs during the first wave of the pandemic, reporting severe symptoms of anxiety (11%) and depression (6%) [18]. Nevertheless, they described the experiences as extreme (13%): “thoughts of being better off dead, or of hurting themselves” [18]. In the first wave of the pandemic, the UK had a high amount of 175,000 COVID-19 patients and 10,000 in need of critical care, causing an extreme workload and lack of resources [18]. In contrast, in Norway, the ICUs had a relatively low total burden of COVID-19 cases during the first wave of the pandemic [3,4]. A higher level of distress in ICU frontline HCPs has been found in other countries: in 21 French ICUs, anxiety symptoms were found in 50% and depression in 30% [15], while 46% reported symptoms of anxiety and depression in a large hospital in Switzerland when they increased the COVID-ICU capacity from 30 to 180 beds [41]. The fact that HCPs in the present study had ICU experience and were prepared to work in COVID-ICU, due to simulation training and sessions of wearing PPE [25], might have influenced the level of anxiety and depression. Other possible factors might be that HCPs, and physicians in particular, experienced their work as meaningful and they receive sufficient support and advice, as in the present study [25].

Health fears were significantly associated with worse mental health outcomes in HCPs during the outbreak as concluded in a review [8]. In the present study, more than half of the respondents were afraid of infecting family, colleagues, patients, or friends, which may be connected to the reported shortage of PPE [23,41]. The fear of being infected and subsequently dying is relevant, as approximately 10% of all COVID-19 infections worldwide have been among HCPs [38], and at the end of 2020, the number of COVID-19 deaths among nurses from 59 countries was 2262, as reported by the International Council of Nurses [42].

Similarly, to the levels of anxiety and depression symptoms, we found low levels of symptom-defined PTSD, and nurses with the highest prevalence rate (7.1%), which may have triggered psychological reactions due to feeling overwhelmed and/or perceiving fear of being infected or infecting their family. This may be related to the high level of education and professional experience [25], but interestingly, a study of the general Norwegian population (n = 4527) during the early stage of the COVID-19 pandemic found PTSD symptoms in 12.5% of men and 19% of women [43], and female gender and lack of social support were two factors associated with PTSD. The latter study [43] used a stricter analysis identifying respondents with findings on the four DSM-V groups identifying PTSD without a cut-off score. Nevertheless, there were fewer cases of symptom-defined PTSD in the ICU HCPs. In countries where the health care workload overwhelmed the local medical resources during the first wave, the prevalence rate for PTSD symptoms was higher: e.g., in UK ICU HCPs it was 40% [18], and in Spain 30.3% [38]. In Switzerland, 22% of the ICU-HCPs (n = 352) reported peritraumatic distress (PDI), i.e., during and immediately after a traumatic event, and risk for PTSD in the pandemic [41]. In China, from Guangdong Province, 10.7% of medical staff included a small group from the respiratory department that had PTSD symptoms (PCL-5 cut-off 33) [44]. The cut-off threshold for identification of PTSD symptoms must be taken into consideration. We used 31 as a cut-off versus 13 in the UK study [18]. However, other studies are using previous PCL versions, and thus, they identified higher symptom levels for probable PTSD cases [45,46]. 

### 4.2. Work Effort, Disturbance of Social Life, and Support

Nurses, physicians, and leaders in our study perceived their individual work effort as satisfying, successful, and meaningful, and this attitude may have protected them against stress reactions [27]. Kramer et al. [27] (n = 3669) found that 43% of German respondents of HCPs in ICUs were professionally satisfied with their work effort and felt appreciated by the management of the hospital, and 91% answered they were positive to continue working at ICUs after the pandemic. This corresponds with our findings that the majority (i.e., 3 out of 4) had not considered quitting, leaving or terminating their job during the pandemic. This steadiness may be due to being experienced ICU-HCPs, being prepared for possible new waves, and having previous work experience under similar conditions in isolation units, while wearing PPE [25,27]. Another reason may be the number of patients hospitalized in COVID-19 ICUs in Norway (n = 217) [4], compared to, for example, the Lombardy region, in Italy (n = 3997) [47]. The moderate number of patients may have strengthened dealing with the total situation of the pandemic in the hospitals, since the workload was not overwhelming for the Norwegian or the Germans HCPs [4,27].

In the present study, the COVID-ICU leaders were significantly more satisfied professionally with their own professional work effort and had a significantly higher mean score compared to physicians and RNs in terms of extending themselves to high work effort. This controversy was unexpected but may have to do with the personal characteristics and preferences of those who choose to become leaders. However, caring for psychosocial well-being in pandemic situations is recommended for both the hospital clinical staff, the general population, and the leaders to prevent distress over time [18,29]. 

Disturbance of social life was experienced during work in cohorts (large areas of many patients) without breaks for a long period of time, but also in private life. We found that 53% (n = 257) felt socially isolated. HCPs in the present study reported suffering from social isolation and reduced possibilities to have social contacts indoors (7%) but reported increased contact via social media (75%). The Norwegians have smartphones and are used to social media. Similar findings from social isolation and fewer contacts are described in literature reviews worldwide [48,49].

During the first wave, the private life resulted in sleep problems and in New York reported 26% on disturbance of sleep (severe or very severe) and an additional 45% reported moderate sleep problems [28]. Some people used alcohol or drugs to reduce the problems. Increased use of alcohol (10%) was reported during the first wave in the present study, a percentage similar to UK ICU staff (7%) [18], but lower compared to the 22% of HCPs in Switzerland [41]. These numbers are not necessarily of concern, as they may reflect that increased use may be transient, subsiding once the pandemic is less prominent. However, personnel should be warned that increased use of alcohol may easily become a problem, especially if they consume alcohol to cope with distress. On the other hand, 25% of respondents consumed less alcohol. We do not know the reason for this, but it is possibly related to decreased social events and social isolation. In addition, the number of HCPs who used sporadic psychoactive medications; for example, sleep inducers (9.9%) and tranquilizers and antidepressants (<1%) were low. In comparison, in France, of 1058 respondents (nurses and physicians), 5% were using psychotropic drugs before the pandemic, and 24% reported starting or increasing use of tobacco, alcohol, cannabis, cocaine, or other drugs during the pandemic [15]. This may indicate that symptoms of anxiety and depression were manageable without medication, either being transient, non-severe, or considered part of a normal reaction.

In the present study, only 5% of the respondents reported follow-up briefings with their leaders after a shift in the ICU. The physicians had perceived sufficient advice and support to a higher degree relative to the other groups, in addition to perceiving the work as meaningful. Another helpful form of support was talking with HCP colleagues at work or of duty, as many of our respondents preferred. In another study, nurse leaders were described as “anchors” in facilitating frontline nurses’ psychological adaptation [45]. In a Cochrane review of frontline workers during pandemics, workplace interventions, such as training, communication, counselling, or psychology service, were reported as beneficial to increasing resilience and mental health, albeit with low evidence [49]. Additionally, again, in Sweden, the implementation of psychological support with peer support, psychologist-supervised and unsupervised group sessions, and manager support was evaluated positively and successfully during work hours [29]. Offering comprehensive support was recommended to manage the ongoing crisis together with preparedness and resilience [6,24,41]. Explicit communication and support from the organization and social support [8], as well as strategically targeted prevention and intervention programs [5,24,29], have been considered as protective factors for PTSD. The implication for policymakers is in line with Binnie et al. [24], to have early recognition of psychological symptoms, and provision of care for the mental health of ICU HCPs. We believe the gaps between wanted and received measures, highlight that support from both leaders and mental health care workers, experts, and therapists have a high potential for improvement and should be addressed in a sustainable response to pandemics. 

### 4.3. Strengths and Limitations

The strength of the present study is the nationwide survey including 27 of the 28 COVID-ICUs in Norway. Another strength is that it includes HCPs from three sub-groups working in the ICU-context. Two internationally recognized and validated checklists, HCL-10 and PCL-5 were part of the questionnaire, and some of the items in the questionnaire were developed by the research group using a modified Delphi method. We also consider the pilot study (n = 5) of the questionnaire package for face validity, reliability, and feasibility as a strength, leading to a minor revision before implementation of the main study. 

A limitation, due to performing a sample calculation ahead of the study, is the lack of a completely national registry of employed HCPs in ICUs. At the present time, the Norwegian directorate of health has initiated a national mapping. On a national level, 96% (27 of 28) of the hospitals with COVID-ICUs participated. The sample size and result will not be representative for all ICUs in the country, and there may be a risk for selection bias, since the head of each hospital ICU decided to whom the invitations would be forwarded. Another limitation of the study is that it includes relatively few physicians. Despite the above limitations, our findings from the first wave are of international interest, and the work will continue with analyses in the longitudinal project. 

## 5. Conclusions

The ICU health care professionals in Norway providing care for COVID-19 patients during the first wave of the pandemic generally had a low level of anxiety, depression, and symptom-defined PTSD. More symptoms were found in the youngest HCP with a few years of work experience. The HCPs experienced isolation in their social life due to the lockdown, in addition to fear of infecting others. Support from colleagues was perceived as the most helpful psychosocial support. The leaders reported a higher mean score than physicians and RNs regarding the experience of extending themselves to a high work effort, which they felt was expected at work in ICUs, from managers and colleagues. 

## Figures and Tables

**Table 1 ijerph-19-07010-t001:** Pre-COVID-19 demographic characteristics of respondents’ professional background in the Norwegian ICUs (N = 484).

Variables	
Age (Range 24–65): Mean (SD)	44.8 (10)
Sex female, n (%)	377 (77.9)
Married or partner, n (%)	362 (74.8)
**PROFESSION, n (%)**	
Registered Nurse (RN)	392 (81.0)
Critical care nurse	305 (75.6)
Nurse anaesthetist	24 (5.4)
Operating room nurse	7 (1.5)
Paediatric nurse	1 (0.02)
Other registered nurse	55(10.7)
Medical doctor (MD)	43 (8.9)
General anaesthesiologist including under	29 (4.9)
ICU specialization	
Anaesthesiologist with ICU specialization	10 (2.9)
Medical doctor of other speciality with ICU	4 (0.08)
specialization	
Leader (RN or MD)	49 (10.1)
**PROFESSIONAL EXPERIENCE**	
Years of professional experience, all participants	
pooled, n (%)	
<1 year	51 (10.5)
1–5 years	136 (28.1)
>5 years	297 (61.4)
MD, Years of professional experience	
(Range 0–35), Mean (SD)	17.4 (9)
RN, Years of professional experience	
(Range 3–42), Mean (SD)	19.6 (9)
Previous intensive care unit experience, n (%)	444 (91.7)

**Table 2 ijerph-19-07010-t002:** Symptoms of anxiety and depression in health care professionals using the Hopkins Symptom Checklist-10 (HSCL-10) during the past week and symptom-defined PTSD (PCL-5) during the past month (N = 484).

	RNs (*n* = 392)	Physicians (*n* = 43)	Leaders (*n* = 49)	*p*
Anxiety and depression HSCL-10				
Symptoms of anxiety * Mean (SD) Symptoms of depression * Mean (SD) HSCL-10 total score, Mean (SD) CI 95% Range (min–max)	1.31 (0.48) 1.41 (0.50) 1.36 (0.46) 1.31–1.41 (1.0–3.6)	1.20 (0.40) 1.31 (0.46) 1.27 (0.42) 1.14–1.40 (1.0–2.5)	1.22 (0.29) 1.27 (0.32) 1.29 (0.33) 1.19–1.38 (1.0–2.2)	0.17 0.13 0.098
Symptoms of anxiety *n* (%) Symptoms of depression *n* (%) No. total symptoms HSCL-10 (%)	42 (10.71) 51 (13.01) 49 (12.5)	5 (11.63) 4 (9.30) 5 (11.63)	2 (4.08) 2 (4.08) 2 (4.08)	0.098
PTSD (PCL-5)				
PCL-5 total score ** Mean (SD) CI 95% Range (min–max)	10.2 (11.02) 9.11–11.30 (0–64)	5.2 (8.16) 2.70–7.72 (0–36)	7.94 (8.23) 5.57–10.30 (0–35)	<0.001 ***
No. total symptom-defined PTSD (%)	28 (7.14)	1 (2.32)	2 (4.08)	<0.001 ***

Note: * Scale HSCL-10 (1 = Not bothered to 4 = Very much bothered), cut-off score of ≥1.85, ** Scale PCL-5 (0 = Not at all to 4= Extremely) cut-off score of ≥31, CI = confidence intervals; SD = standard deviation, Kruskal–Wallis test used between groups, *** pair-wise comparison with Bonferroni correction detected no difference between groups tests.

**Table 3 ijerph-19-07010-t003:** Results of multivariate quantile regression analysis of factors predicting probable mental health disorder (N = 56) (Dependent variable HSCL-10 ≥ 1.85).

Exploratory Variables Estimates (q = 0.5) R^2^ = 0.160	Coefficient	Standard Error	95%CI	t	*p-Value*	*p-Adj*
Age	−0.16	0.0063	−0.028, −0.003	−2.464	0.182	0.017
Previous work experience (<5 years)	0.400	0.1509	0.097, 0.703	2.650	0.018	0.011

Note: CI = Confidence intervals, *p*-*adj* = *p value* adjusted for multiple testing, R^2^ = Pseudo R Squared.

**Table 4 ijerph-19-07010-t004:** Work effort, with subsequent professional satisfaction and experienced proficiency in COVID-ICU Health care professionals, mean (SD), N = 484.

	RNs (n = 392)	Physicians (n = 43)	Leaders (n = 49)	*p*
To what extent * are you satisfied professionally with the following aspects of your work?				
Your own work efforts	3.86 (0.03)	3.77 (0.09)	3.92 (0.09)	0.42
Your closest partner’s work efforts	3.99 (0.03)	4.09 (0.09)	4.12 (0.08)	0.27
Your department’s work efforts	3.92 (0.04)	4.19 (0.11)	4.16 (0.08)	0.04 ***
Your organization’s work efforts	3.40 (0.05)	3.56 (0.15)	3.35 (0.10)	0.30
The responsible authorities’ work efforts	3.64 (0.04)	3.49 (0.13)	3.53 (0.11)	0.43
To what extent * have you experienced proficiency during work in a COVID-19 ICU?				
The work was a success	4.0 (0.03)	3.98 (0.08)	3.94 (0.08)	0.75
The work was meaningful	4.12 (0.04)	4.44 (0.1)	4.12 (0.08)	0.009 ****
I received sufficient advice and support	3.24 (0.04)	3.74 (0.09)	3.22 (0.11)	<0.001 ****
I fell short of the work tasks to be performed **	2.54 (0.04)	2.26 (012)	2.49 (0.09)	0.10
I was impeded in the work (e.g., by a lack of PPE, by the bureaucracy) **	2.48 (0.05)	2.51 (0.17)	2.73 (0.13)	0.18
Did you overextend yourself as a large effort was expected from your team, managers or colleagues? **	2.98 (0.06)	2.51 (0.15)	3.45 (0.15)	<0.001 ****

Note: * Scale (1 = not at all to 5 = to a very high degree), ** Negative question/answer, PPE = personal protective equipment, Kruskal–Wallis test were used between groups, *** pair-wise comparison with Bonferroni correction detected no differences between groups for multiple tests, **** pair-wise comparison with Bonferroni correction detected differences between groups tests.

**Table 5 ijerph-19-07010-t005:** Perceived recognition and support during the pandemic in health care professionals (mean [SD], N = 484).

	RNs (n = 392)	Physicians (n = 43)	Leaders (n = 49)	*p*
Recognition and support * during the pandemic				
From the hospital	3.00 (0.05)	3.44 (0.12)	3.24 (0.12)	0.005 **
From family or friends	3.98 (0.04)	3.79 (0.12)	3.92 (0.10)	0.27
From the responsible authorities	3.03 (0.05)	3.09 (0.12)	3.29 (0.15)	0.22

Note: * Scale (1 = not at all to 5 = to a very high degree), statistical test was Kruskal–Wallis test, ** pair-wise comparison with Bonferroni correction detected a significant (0.009) difference between RNs and Physicians.

## Data Availability

Data supporting reported results can be requested to an additional, non-author, Johannes Lagethon Bjørnstad, Department of Cardiothoracic surgery, j.l.bjornstad@medisin.uio.no. There are not publicly archived datasets analysed or generated during the study.

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
