# Peer review of "Symptoms of Anxiety, Depression, and Post-Traumatic Stress Disorder in Health Care Personnel in Norwegian ICUs during the First Wave of the COVID-19 Pandemic, a Prospective, Observational Cross-Sectional Study"

_ijerph, 2022, doi:10.3390/ijerph19127010_

Round 1
Reviewer 1 Report
The Introduction provided sufficient backgrounds and had relevant references.
Methods are very well described. There was some concerns about methodology and generalizability but the authors clearly presented that in the study limitations.
Results were clearly presented and discussion follows the results. Author could add some implications for policymakers in the Discussion on the basis this results and results of another studies.
Author Response
Response to reviewers
We thank the reviewers for valuable advice and suggestions. Please find our detailed responses as follows and citation/changes made in the manuscript with is in italic “xxxxx”.
Reviewer 1
Comments and Suggestions for Authors
The Introduction provided sufficient backgrounds and had relevant references.
Methods are very well described. There was some concerns about methodology and generalizability but the authors clearly presented that in the study limitations.
- Results were clearly presented and discussion follows the results. Author could add some implications for policymakers in the Discussion on the basis this results and results of another studies.
Response R1:1). We have added on page 12, line 482-484: “The implication for policymakers is in line with Binnie et al [24], to have early recognition of psychological symptoms, and provision of care for ICU HCP mental health.”

Reviewer 2 Report
Dear authors
Thank you for this great contribution.
I detail some minor revisions and attach the manuscript with comments.
Review all decimal numbers, they must be separated by "." and not by "," (For example, put in the line 190 "p = 0.05 and remove p = 0.05").
Report the exact data of the value of "p" always (For example, detail this information in the last two rows of table 2). Review the entire manuscript and provide this information where it is not fully detailed.
Kind regards.

Author Response
Reviewer 2:
- Introduction, 1.2. Detail bibliographic citation that supports this statement.
Response R2:1 Introduction. On page 3 line 100 we have added a bibliographic citation “[3]” to The Norwegian Government’s Management of the Coronavirus Pandemic.
- Methods, 2.1 Was it sent to the participants' email? In that case, how were the email addresses obtained?
Response R2: 2 Methods. Thank you for a valuable comment to clarify how the contact and the email addresses were obtained. The local leader redistributed the invitation to the employees in their ICU with the study information and a direct link to participate in the study, by first sign the informed consent sheet, and then answer the questionnaire. We, the researchers did not obtain the email addresses. We have added on page 4 line 147:” The HCP could voluntary follow a direct link for electronic signature to the informed consent and could continue to respond the study questionnaire.”
- Results, 3.1 Did all the people to whom the questionnaire was sent answer? What was the response rate? Was the reason for the non-response known?
Response R2: 3 (Results). We have no information regarding the response rate since the leaders did not report the number of employees receiving the invitation to participate. However, we reach 96% of all ICUs caring for COVID-19 patients in the country. This information is added on page 12 line 499: “On a national level 96% (27 of 28) of the hospitals with COVID-ICUs participated”.
4.(i.e. 1-6) Discussion. It would be convenient to include in the discussion what the levels of the variables studied were like before the pandemic.
Response R2: 4 1). In our country, no study was performed before the pandemic on the variables of for example anxiety, depression, or post-traumatic stress disorder. However, we asked the respondents if they have had previous symptoms of anxiety and depression. The result is added as suggested: page 10 line 363 “Approximately one-sixth of HCP reported previous symptoms of anxiety and depression (16.1% and 20.0%, respectively).”
2 (4 Discussion): younger age, and limited work experiences (<5 years) significantly increased the symptoms of anxiety and depression. Comment This needs to be discussed and compared to other studies.
Response R2: 4 2. We have added two references on page 10 line 368-371: “To be younger HCP in our study, was a comparable risk factor for symptoms and the study from Metha et al [39] could confirm the results. However, the risk factor of being less experienced HCP was not confirmed in the Canadian study. Less experienced nurses were found by Kramer et al [27] to experience psychological distress.”
3 (4 Discussion): which may have protected them against stress reactions. This statement must be supported with a bibliographic citation.
Response R2 3. On page 10 line 355, we have inserted [25].
4 (4 Discussion): Greenberg et al. (2021) underline in their UK study 326 (n=709) [18]. Comment Remove the year in parentheses and put its place the number which this bibliographical citation corresponds.
Response R2: 4. On page 10 line 366, we have removed (2021) and it is in reference [18].
5 (4 Discussion). The prevalence of anxiety and depression symptoms in the present study was low compared to other studies, but comparable to UK HCPs working in ICUs during the first wave of the pandemic, reporting severe symptoms of anxiety (11%) and depression (6%) [18]. Comments: List bibliographic citation of these studies.
Response R2: 5 We have added the review by Muller et al from 2020 describing prevalence of anxiety and depression symptoms. On page 10 line 375 as reference “[8]”.
6 (4 Discussion): Other possible factors might be that HCPs, and physicians in particular, experienced their work as meaningful and they receive sufficient support and advice. This statement must be supported with a bibliographic citation.
Response R2 4 6. Well, this was result from present study. We have added on page 10 line 388-390: “Other possible factors might be that HCPs, and physicians in particular, experienced their work as meaningful and they receive sufficient support and advice, as in present study [25].”

Reviewer 3 Report
In my opinion, this is a well-thought out and well-executed investigation. The topic is no longer new (the first wave of Covid is far behind us), but it provides useful information to understand what happened in Norway and to compare it with what happened in other countries.
The introduction focuses well the topic of study. Aspects for improvement: It would be necessary to include another section that contextualizes the framework in Norway in the first wave in relation to Covid. Specifically, the impact on society and on the healthcare system. Some elements that occurred in other countries have been reflected in the introduction, but there is no contextualization of what happened in Norway. For example, the existence of lockdowns in the country or restrictions on social interaction in specific geographical areas, whether the healthcare system or healthcare personnel were overwhelmed or not, whether it was necessary to create COVID-ICUs in addition to the existing ICUs in hospitals, the number of personnel being asked to work outside one's expertise area, the availability of material and personal protective equipment, etc... I think this framework is key for the reader to be able to understand and interpret the results.
The methodology describes all the necessary elements. Validated tests have been used. All ethical requirements have been met. Aspects for improvement: 28 eligible hospitals are mentioned, of which 27 participated. It should be indicated what percentage these hospitals represent with respect to the total number of hospitals in the country, and if not 100%, what was the criterion for deciding that they were eligible. The Kruskall-Wallis test was used, but no mention is made of the post-hoc contrasts that would have to be performed between pairs when statistically significant differences were detected.
The results are complete and clear. The tables are well constructed. Aspects for improvement: The results of the post-hoc contrasts should be included in the corresponding tables.
The discussion compares the results with the relevant literature. The strengths and limitations of the study are included. Aspects for improvement: It is mentioned as a limitation not being able to calculate the sample size because there is no specific registry at the national level. Providing the data on the % of ICUs that participated in the study would moderate this limitation.
Author Response
Reviewer 3:
In my opinion, this is a well-thought out and well-executed investigation. The topic is no longer new (the first wave of Covid is far behind us), but it provides useful information to understand what happened in Norway and to compare it with what happened in other countries.
1.The introduction focuses well the topic of study. Aspects for improvement: It would be necessary to include another section that contextualizes the framework in Norway in the first wave in relation to Covid. Specifically, the impact on society and on the healthcare system. Some elements that occurred in other countries have been reflected in the introduction, but there is no contextualization of what happened in Norway. For example, the existence of lockdowns in the country or restrictions on social interaction in specific geographical areas, whether the healthcare system or healthcare personnel were overwhelmed or not, whether it was necessary to create COVID-ICUs in addition to the existing ICUs in hospitals, the number of personnel being asked to work outside one's expertise area, the availability of material and personal protective equipment, etc... I think this framework is key for the reader to be able to understand and interpret the results.
Response R3: 1. Thank you for the comments. We have addressed the framework in Norway on page 2 line 50-57. “In Norway, the first wave of the pandemic was characterized by geographic variation in cases. South-eastern region of Norway had large outbreaks while other regions saw few cases. The government implemented restrictions and stringent lockdown i.e. to stay in municipality of residence. People were encouraged to work from home office, and to minimize social contact outside the family. Schools were closed, and hospital visits were restricted. It was necessary to “build” COVID-ICUs and wards to special care for the patients were created. Personnel were asked to work in new areas and wards, also with lack of personal protective equipment and increased risk of infection [3].”
2.The methodology describes all the necessary elements. Validated tests have been used. All ethical requirements have been met. Aspects for improvement: 28 eligible hospitals are mentioned, of which 27 participated. It should be indicated what percentage these hospitals represent with respect to the total number of hospitals in the country, and if not 100%, what was the criterion for deciding that they were eligible.
Response R3: 2. We have rewritten page 3 line 142: ”Respondents were defined as frontline HCP actually caring for COVID-19 patients centralized in 28 hospitals, satisfying the criterion of having a COVID-ICU.”
- The Kruskall-Wallis test was used, but no mention is made of the post-hoc contrasts that would have to be performed between pairs when statistically significant differences were detected.
Response R3: 3. Yes, we have now identified the between pairs of groups: Pair-wise correction with Bonferroni multiple tests are added in table 2, 4 and 5. For example *** pair-wise comparison with Bonferroni correction detected no differences between groups for multiple tests, **** pair-wise comparison with Bonferroni correction detected differences between groups tests. And results from significant tests in Table 4 of significance is further described in the text. Page 8 line 311-318, “Moreover, the work was experienced as meaningful and Kruskal-Wallis test (after correction) revealed a significant) difference between leaders compared to physicians (0.012) and RNs (0.002) compared to physicians. In receiving sufficient advice and support, significant differences were detected between the leaders (0.03) and physicians and between RNs (p<0.001) and physicians. The variable overextending i.e. in a large effort was expected from the team, managers or colleagues results showed significant differences between groups physicians (0.012) to RNs, versus leaders (p<0.001).”
- The results are complete and clear. The tables are well constructed. Aspects for improvement: The results of the post-hoc contrasts should be included in the corresponding tables.
Response R3: 4. Please, see R3: 3 response.
5.The discussion compares the results with the relevant literature. The strengths and limitations of the study are included. Aspects for improvement: It is mentioned as a limitation not being able to calculate the sample size because there is no specific registry at the national level. Providing the data on the % of ICUs that participated in the study would moderate this limitation.
Response R4: 5. We have added page 12 line 500: “On a national level 96% (27 of 28) of the hospitals with COVID-ICUs participated”.
